# Influence of Thickness and Sputtering Pressure on Electrical Resistivity and Elastic Wave Propagation in Oriented Columnar Tungsten Thin Films

**DOI:** 10.3390/nano10010081

**Published:** 2020-01-01

**Authors:** Asma Chargui, Raya El Beainou, Alexis Mosset, Sébastien Euphrasie, Valérie Potin, Pascal Vairac, Nicolas Martin

**Affiliations:** 1Institut FEMTO-ST, UMR 6174, CNRS, ENSMM, Univ. Bourgogne Franche-Comté, 15B, Avenue des Montboucons, 25030 BESANCON CEDEX, France; asma.chargui@femto-st.fr (A.C.); raya.elbeainou@femto-st.fr (R.E.B.); alexis.mosset@femto-st.fr (A.M.); sebastien.euphrasie@femto-st.fr (S.E.); pascal.vairac@femto-st.fr (P.V.); 2Laboratoire Interdisciplinaire Carnot de Bourgogne (ICB), UMR 6303, CNRS, Univ. Bourgogne Franche-Comté, 9, Avenue Alain Savary, BP 47 870, F-21078 DIJON CEDEX, France; Valerie.Potin@u-bourgogne.fr

**Keywords:** sputtering, GLAD, tilted columns, anisotropy, electrical resistivity, elastic wave propagation

## Abstract

Tungsten films were prepared by DC magnetron sputtering using glancing angle deposition with a constant deposition angle α = 80°. A first series of films was obtained at a constant pressure of 4.0 × 10^−3^ mbar with the films’ thickness increasing from 50 to 1000 nm. A second series was produced with a constant thickness of 400 nm, whereas the pressure was gradually changed from 2.5 × 10^−3^ to 15 × 10^−3^ mbar. The A15 β phase exhibiting a poor crystallinity was favored at high pressure and for the thinner films, whereas the bcc α phase prevailed at low pressure and for the thicker ones. The tilt angle of the columnar microstructure and fanning of their cross-section were tuned as a function of the pressure and film thickness. Electrical resistivity and surface elastic wave velocity exhibited the highest anisotropic behaviors for the thickest films and the lowest pressure. These asymmetric electrical and elastic properties were directly connected to the anisotropic structural characteristics of tungsten films. They became particularly significant for thicknesses higher than 450 nm and when sputtered particles were mainly ballistic (low pressures). Electronic transport properties, as well as elastic wave propagation, are discussed considering the porous architecture changes vs. film thickness and pressure.

## 1. Introduction

Among the solid-state physical properties, understanding the propagation of electrons and elastic waves in materials remains a challenging task. It was commonly pointed out that the control of transport mechanisms in solids greatly influences resulting applications in devices used as bulk and surface acoustic wave systems (BAW and SAW), semiconductors, gas sensors, and so on [1,2,3]. It is also admitted that the crystal symmetries mainly affect the direction-dependent properties of materials due to an intrinsic anisotropy at the atomic scale coming from spatial differences between bonds in the crystal structure. Such anisotropy becomes negligible or even vanishes when the material becomes polycrystalline, which is particularly true when the material dimension changes from 3D to 2D. As a result of the loss of this intrinsic anisotropy, structuring at the micro- and nanoscale appears as an attractive strategy to produce directional behaviors in surfaces and thin solid films [4,5,6,7]. These spatially organized surfaces and films may exhibit asymmetric characteristics in electronic and ionic transports, electromagnetic wave propagations, magnetic properties, or even mechanical and tribological performances [8,9,10].

For the last decades, tuning the physical properties of thin films and surfaces and producing anisotropic behaviors have attracted many researchers since it allows producing multifunctionality within the same material [11,12]. Then, the structuring of thin solid films at the micro and nanoscale using top-down or bottom-up strategies has become a pertinent tool to get original patterns and designs favoring anisotropic behaviors. Among these strategies, growth of thin films by vacuum processes, such as magnetron sputtering by the GLAD technique (GLancing Angle Deposition), recently became a very motivating way to create original architectures (tilted columns, zigzags, spirals, etc. [13,14,15,16]) and thus, to develop anisotropic properties [17,18,19]. However, some scientific and technological challenges are still to be addressed, such as the tunability of anisotropic characteristics and understanding the correlations between created structures and propagation of waves, electrons, ions, or temperature through these structured thin films. In addition, growth conditions by sputtering (pressure, temperature, gases, etc.) may strongly influence some characteristics and the final structure of as-deposited thin films. These experimental parameters restrain some achievable properties of thin films, and they still need to be explored, particularly in the GLAD technique where the growth mechanisms may largely differ compared to conventional sputtering [20,21].

In this article, we report on tilted columnar tungsten thin films sputter-deposited by magnetron sputtering using the GLAD technique with a constant deposition angle of α = 80°. Two deposition parameters have been studied: the film’s thickness and the argon sputtering pressure giving rise to two series of films, i.e., a first series with constant pressure and a growing film thickness, and a second series with a constant thickness and different pressures. Electrical resistivity and surface elastic wave propagation have been systematically investigated for both series. This study is motivated by the understanding of anisotropic behaviors in terms of electronic transport properties and elastic wave propagation in nanostructured thin films exhibiting a tilted columnar architecture. The choice of tungsten films is due to its ability to produce a columnar structure with tunable cross-section morphologies depending on deposition time and sputtering conditions. By increasing the film’s thickness, it has been shown that the electrical resistivity and elastic wave velocities were both reduced, whereas anisotropy was favored. A higher pressure produced more resistive films with an increase in the elastic wave velocity, where anisotropic behaviors were reduced. The evolution of these physical properties as a function of the film’s thickness and argon sputtering pressure is discussed, taking into account the evolution of the crystallographic structure and the films’ morphology at the micro and nanoscale.

## 2. Materials and Methods

Tungsten films were sputter-deposited on glass and (100) Si substrate by DC magnetron sputtering from a pure metallic target (51 mm diameter and 99.9 at.% purity) in a home-made vacuum chamber. The experimental deposition system was a 40 L sputtering chamber pumped down via a turbo-molecular pump backed by a primary pump leading to a residual vacuum of 10^−8^ mbar. The target current was fixed at 100 mA, and the target-to-substrate distance was 65 mm. No external heating was applied during the growth stage, and depositions were carried out at room temperature. The GLAD (GLancing Angle Deposition) technique [22] was implemented to produce tilted columnar architectures. It consists of depositing thin films under conditions of obliquely incident flux of the sputtered particles and on a fixed or mobile (rotating) substrate. In this study, the substrate is tilted at an angle α = 80° without rotating (fixed substrate). Two series of samples were produced. For the first series, an argon flow rate of 5.6 sccm and a constant pumping speed of 24 L s^−1^ were used. These conditions produced an argon sputtering pressure of 4.0 × 10^−3^ mbar, and the deposition time was systematically changed to get a film thickness variation from 50 to 1000 nm. For the second series, the deposition time was set to deposit around 400 nm thick films changing the argon sputtering pressure from 2.5 × 10^−3^ up to 15 × 10^−3^ mbar (pumping speed and argon flow rate were both adjusted to get the required range of pressures).

The crystallographic structure of W films was characterized by X-ray diffraction (XRD). Measurements were carried out using a Bruker D8 focus diffractometer with a Cobalt X-ray tube (Co λ_Kα1_ = 0.178897 nm) in a *θ*/2*θ* configuration. Patterns were recorded with a step of 0.02° per 0.2 s and a 2*θ* angle ranging from 20° to 80°. Scanning electron microscopy (SEM) was used to view the surface and the fractured cross-section of the films with a JEOL JSM 7800 field emission SEM. DC electrical resistivity of the films was measured at room temperature in air by the four-probe van der Pauw method. The measurements were carried out on glass substrates, and the surface anisotropic electrical resistivity was determined using the method previously developed by Bierwagen et al. [23].

A femtosecond pump-probe setup was used for the measurement of the elastic wave propagation. This technique is based on an ultrashort laser pump pulse that interacts with the surface of the sample. This leads to a sudden temperature rise that will excite the surface into vibration through the thermoelastic effect. The subsequent reflectivity modifications are measured by a low-power second laser probe pulse, with an energy around a tenth of the pump pulse energy. Two femtosecond Ytterbium lasers (T-Pulse Duo from Amplitude System were used in our setup with a heterodyne configuration also called asynchronous optical sampling technique (ASOPS), where an increasing delay between the pump and probe pulses was induced by a small frequency shift of their repetition rates [24]. This frequency shift was 700 Hz with a repetition rate of 48 MHz. This gave rise to an additional time difference between the pump and the probe beam of around 300 ps every pulse. The whole time spanning 21 ns was thus obtained with a resolution of about 1 ps. The average pump and probe beam powers were 5 and 0.5 mW, respectively. Both beams were focused on the sample with a 1 μm spot radius. Thanks to a lens mounted on a 2D translation stage in the pump optical path, the pump-probe distance could be scanned to image the reflectivity. From this imaging, we deduced the dispersion curves obtained by computing the 2D-FFT (2 Dimensions Fast Fourier Transform) of the relative reflectivity vs. pump-probe distance and time. The elastic wave group velocities were obtained from the local slope of the dispersion curves (frequency vs. wave number/2π). They were calculated at the most intense normalized reflectivity of the pseudo-Rayleigh mode, which was at the wave number *k* = 2π × 3 × 10^5^ m^−1^ [24].

We used a 3D finite element (FE) model to correlate the porosity with the measured dispersion curves; finite element analyses were performed using COMSOL Multiphysics. The simulation model was based on a silicon block on which lay a nanoporous tungsten film with periodic parallelepipedic holes to model the macroporosity (visible holes in the SEM pictures). The structure was simulated to be infinitely periodic with Bloch–Floquet conditions, both in *x* and *y*-directions. This model is very simplistic but gives the overall tendencies and is a computer-based tool, which does not involve a strong memory requirement.

Figure 1 shows the unit cell assumed in this paper.

It consisted of a square cuboid silicon block (parallelepiped with two opposite square faces) with a side *a* = 500 nm and height *h*_Si_ = 25 μm (not on scale in Figure 1), on which lay a porous tungsten film. A hole with a rectangular base (length *c* and width *d*) was dug in the tungsten film. The side *a* (period) was chosen from the SEM pictures. A previous study [24] showed that the anisotropy is connected to the geometry of the holes, especially their form factor and the choice of the period, the column tilt angle having only a limited influence. The material constants used in the simulations were from literature data [25] for the silicon and modified for the tungsten to take into account the nanoporosity.

Since Biot’s theory [26] for porous solids, several groups have studied the porosity as a key parameter to establish a theory about elastic wave propagation in a system composed of a porous elastic solid. Investigations focused on doped porous Si wafers [27,28] propose an empirical relation between the longitudinal and shear waves (bulk waves) and the porosity following:(1)$$v=v_{0}{\left(1-p\right)}^{k},$$
where *v* is the velocity of the wave in the porous material (m s^−1^), *v*_0_ the velocity of the bulk wave in the bulk material (m s^−1^), *p* the porosity of the material and *k* a parameter, close to one, depending on the nature of the wave, the porosity, and the material. This form of velocity dependence on porosity was used to fit our experimental results with *k* = 1. Neglecting the density of air compared to the density of as-deposited film, the density *ρ* (kg m^−3^) of the nanoporous film is given by:(2)$$\rho =\left(1-p\right)\times \rho_{0},$$
where *ρ*_0_ (kg m^−3^) is the density of bulk material. Thus, the elastic constants vary with nearly the 3rd power of (1 − *p*) [29].

The lattice constant *a* of the unit cell was chosen from the SEM observations, the size of the hole (*b* and *c* parameters), and the nanoporosity *p* were adjusted to fit the experimental data. The global porosity *π* (including the nanoporosity *p* and the macroporosity) is given by:(3)$$\pi =p+\left(1-p\right)bc/a^{2}$$

## 3. Results and Discussion

### 3.1. First Series: Thickness from 50 to 1000 nm

#### 3.1.1. Morphology and Structure

SEM observations of as-deposited W films exhibited a gradual change in surface and cross-section morphologies as a function of the film’s thickness (Figure 2). For the thinnest film (50 nm), no clear shapes could be distinguished from the top view, whereas very small columns (about a few tens nm width) could be seen from the cross-section view (Figure 2a). Despite the very low film thickness, they were oriented in the direction of the incoming particle flux. For this first series, the sputtering pressure was 4.0 × 10^−3^ mbar, which mostly produced ballistic sputtered particles. The incident vapor was thus highly directional, and the shadowing effect became effective from the early growing stage, i.e., after a thickness of a few nanometers. Randomly distributed islands were observed when the thickness was over 100 nm, and tilted columns (*β* close to 39° ± 2°) were even more distinct (Figure 2b). A further increase in thickness from 200 to 1000 nm led to a better-defined microstructure (Figure 2c–f). Top views showed a more and more corrugated surface as the thickness increases. Asymmetric and elongated voids alternated with columns exhibiting an elliptical cross-section following the direction perpendicular to the particle flux, i.e., *y*-direction. This anisotropic microstructure has ever been reported for tungsten and other metallic thin films prepared by GLAD [30]. This column fanning was closely connected to the film growth. During the first growing stages, nuclei were randomly distributed on the substrate surface with no clear structure. As the deposition progresses, the column features gradually fanned out along the *y*-axis (normal to the particle flux) and the shadowing phenomenon started acting. As a result, a transverse growth was favored leading to a well-defined elliptical shape of the columns section and a rising column interspacing with a more voided structure.

Cross-section observations of slanted columnar W structures also showed a continuous evolution as a function of the film’s thickness. Some columns fell under the shadow of bigger ones and become extinct (especially noticeable from the cross-section views of the thickest films in Figure 2e,f). This shadowing-induced competition between columns continuously occurred during the film growth. It was previously shown that the GLAD films morphology is scale-invariant with a power-law scaling connecting thickness and columns’ width in *x*- and *y*-directions [31]. Surface self-diffusion, atomic mobility, ballistic character of incoming particles, or dragging phenomenon are quite a few mechanisms which significantly influence the column broadening and tilting vs. film thickness, and thus, the final structural morphology and anisotropy [32]. Despite several growth simulations in agreement with experimental data and predicting column angle and broadening, the structural anisotropy of GLAD films still remains strongly dependent on operating conditions and experimental methods (cf. Section 3.2 on the role of the argon sputtering pressure).

XRD measurements also showed that the crystalline structure of W GLAD films was meaningfully affected by the film’s thickness, as reported in Figure 3.

For the smallest thickness (50 nm), no significant diffracted signals were recorded but only a broad and poorly intense band located at 2*θ* angle close to 47°, which is related to the (210) planes of the metastable A15 β phase. As the film’s thickness reached 200 nm, more intense peaks were clearly measured and are once again assigned to the β phase. Peaks became even more intense and narrow for thicknesses higher than 450 nm. The bcc α phase appeared with an important diffracted signal corresponding to (110) planes recorded at 2*θ* = 47.12°. Thus, increasing the film’s thickness favors the crystallinity of α and β phases in the columnar structure. From α (110) and β (200) peaks, the crystal size determined using the Scherrer formula reached 31 and 57 nm for α and β phases, respectively, as the thickness reached 1000 nm. In addition to the α phase occurrence, the β phase grew with a preferential orientation along (200) direction. As a result, a growth competition occurred between these two W phases as a function of the film’s thickness. These results can be connected to former investigations focused on the crystallographic structure of W films prepared by conventional sputtering (i.e., deposition angle α = 0°) [33]. During the early film deposition stage, β phase nuclei are initially formed up to a so-called critical thickness, and a phase mixture is produced (appearance of the α phase). Such a critical thickness is in the order of a few to several tens nanometers and largely depends on the experimental growth parameters, especially pressure, deposition rate, and power [34]. Depositing above this thickness by a conventional sputtering process commonly leads to a β to α phase transformation due to a diffusion-controlled process of W atoms [35]. In the GLAD sputtering technique, growth mechanisms may differ since the direction of the W particle flux can yield dense or fibrous morphologies, as the column apexes are in front of the flux or in the shadowing zone [36]. This inhomogeneous growing evolution in the columnar growth significantly produces a completely different microstructural morphology and, thus, different crystallographic structure and properties through the cross-section of a given column. Taking into account that the β phase is favored when the energy of deposited species is reduced [37], one can expect the β phase occurrence on the column sides located in the shadowing zone, whereas the α phase prevails in front of the incoming particle flux. As a result, for thicknesses higher than 200 nm, α and β phases coexisted in W GLAD thin films sputter-deposited with our operating conditions. In addition to the β (200) peak located at 2*θ* = 41.58°, this phase mixture is well illustrated by the shouldered peak close to 47° indicating the presence of both phases. The latter asymmetric signal can be deconvoluted to the β (210) and α (110) Bragg peaks (2*θ* = 46.76° and 47.12°, respectively) to get approximately α and β phase content in the film from the selected phase peak area to the total peak area ratio [37]. For the lowest thicknesses (< 200 nm), films were mainly composed of the β phase, and the α phase started growing at 200 nm (α content was a few percentages by volume). When the film’s thickness reached 1000 nm, the α phase significantly rose with a proportion by volume higher than 40%.

#### 3.1.2. Electrical Resistivity

DC electrical resistivity of GLAD W thin films deposited on glass substrate also varied as a function of the film’s thickness from 1.1 × 10^−5^ to 5.5 × 10^−6^ Ω m, as shown in Figure 4.

It was higher than the bulk value (*ρ*_300K_ = 5.4 × 10^−8^ Ω m for bulk W [38]), which is classically reported in GLAD films [39]. This higher resistivity is assigned to the enhancement of the electron scattering by surfaces and grain boundaries induced by a much more porous structure promoted as the incident angle α tends to 90°. However, the resistivity evolution of W GLAD films did not exhibit the classical saturation effect as typically observed in conventional sputtered films when the thickness exceeds a few tens nanometers [40]. This continuous reduction of resistivity vs. thickness has to be connected to XRD results, which simultaneously showed an increase in the crystal size and favoring of the α phase as a function of the film’s thickness. Since the electrical conductivity of metallic thin films is limited by the scattering of electrons at grain boundaries, an increase in the crystal size induces a longer electron mean free path and thus, improves the electrical conductivity of the films. In addition, Petroff and Reed [41] previously showed that the amounts of α and β phases present in W thin films strongly affect the resistivity because bulk α and β phases exhibit a significant difference of resistivity with *ρ*_α_ = 5.4 × 10^−8^ Ω m lower than *ρ*_β_ = 1.5–3.5 × 10^−6^ Ω m at 300 K [42]). As a result, an increase in the crystal size associated with a larger proportion of the α phase as thickness increased, both contributed to the drop in the films’ resistivity.

Figure 4 also illustrates the influence of the films’ thickness on anisotropic resistivity *A*_ρ_ defined as the resistivity ratio following *x* and *y*-directions (i.e., parallel and perpendicular to the direction of the particle flux). For the lowest thickness of 50 nm, *A**_ρ_* = 1.8, which was surprisingly high since no clear anisotropic morphology was viewed from SEM observations in Figure 2a. On the other hand, some studies have deservedly reported that the shadowing effect may become effective from the first growing stage of oblique angle deposition [43], especially at low pressure for the sputtering process and for deposition angles *α* higher than 80°, which were our operating conditions. It is also worth noting that the first nuclei created nanoscale topographies, thus inducing an initial surface roughness, which is a principal requirement for the shadowing effect to begin. The latter is a key parameter, particularly when the surface diffusion of incoming particles is limited. Such is the case of W atoms where the surface self-diffusion energy *E_d_* = 3.10 eV, which is high compared to other metals where *E_d_* is lower than 1 eV [44]. Thus, atoms such as W are unable to fill-in the shadowed regions (shadowing prevents structural broadening of the columns in the direction of the particle flux). As a result, despite no clear surface structural asymmetry being clearly distinguished from SEM images of 50 nm thick films, a structural anisotropy certainly occurred shortly after the first growing stage.

Anisotropic resistivity became more and more relevant as the film’s thickness increased and reached 2.3 at 600 nm. Although the film’s thickness increased up to 1000 nm, *A**_ρ_* remained higher than 2.2. This anisotropic behavior, which was more obvious for thicknesses higher than a few hundred nm, agrees with other studies focused on the electronic transport properties of GLAD thin films [45]. Such behavior is mainly assigned to the elliptical shape of the columnar cross-section (fanning mechanism during the growth, as shown in Figure 2), being especially prominent in W GLAD thin films [46]. An alternation of dense and voided architecture is rather produced in the direction of the particle flux, whereas dense and chained columns were obtained following the normal direction. A further increase in the film’s thickness did not enhance anisotropic resistivity. This saturation can be associated with a nearly stable crystallographic structure (α and β phase mixture) and nearly unchanged morphology (top and cross-section views by SEM rather show a scaled architecture) from 450 to 1000 nm.

#### 3.1.3. Elastic Wave Propagation

The group velocities of the Rayleigh waves following the *x*- (parallel to the incoming flux) and *y*-direction (perpendicular to the incoming particle flux) were obtained from the local slope of the dispersion curves for a wave vector *k*/2π = 1/*λ* = 3 × 10^5^ m^−1^ in our case (where *λ* is the wavelength). The calculated group velocities of the Rayleigh waves of W thin films change as a function of the film’s thickness, as shown in Figure 5.

As the film’s thickness increased, velocities decreased, and the anisotropy coefficient *A_v_*, defined as the velocity ratio following *x* and *y*-directions, increased. Since the films were smaller than the wavelength, the Rayleigh waves also propagated into the substrate. Therefore, one can expect an increase in *A_v_* with the thickness. Moreover, as the Rayleigh velocity in silicon (4917 m s^−1^ along the <100> direction [47]) is higher than in tungsten (2646 m s^−1^ [47]), the decrease in velocities is also expected. However, analytical calculation with a homogeneous film and simulations showed that the effect of the substrate is not enough to explain these behaviors (cf. the computing of porosity at the end of this section).

For the smallest thickness (50 nm), velocities were similar along *x* and *y*-axes (3450 ± 50 m s^−1^ and 3630 ± 50 m s^−1^, respectively). As a result, the anisotropic coefficient *A_v_* equaled to 1.05, contrary to the resistivity anisotropy (*A**_ρ_* = 1.8). This result could be expected since a good proportion of the waves propagated into the substrate. It is worth noting that these velocities were higher than the surface wave velocities of tungsten bulk metals, a consequence of the higher velocity in Si.

Increasing the thickness up to 450 nm led to an important drop for the pseudo-Rayleigh wave velocity with *v_x_* = 1100 m s^−1^ and *v_y_* = 1848 m s^−1^ along *x*- and *y*-directions, respectively. This sudden decrease in velocities is mainly attributed to the formation of voided microstructure and large spaces between the inclined columns, which became more and more important as the thickness increased. Moreover, the column features gradually fanned out along the *y*-direction while shadowing prevented significant structural broadening along the *x*-direction. The cross-section thus became increasingly elliptical as deposition continued, and the basic columnar microstructure exhibited structural anisotropy because of the development of an elongated column cross-section. Therefore, the significant difference between velocities *v_x_* and *v_y_* is linked to the structural anisotropy, which developed in the film plane. Increasing the thickness to and above 800 nm led to a saturation of the anisotropic coefficient *A_v_* tending to 2.0. This stabilization, as with the resistivity, comes from the stable crystallographic structure (α and β phase mixture) and the nearly unchanged and scalable morphology.

To separate the effect of the substrate from that of the film’s morphology, FE simulations, including macro and nanoporosities, were performed with a simplistic model, as described in Section 2. Table 1 illustrates the evolution of the global porosity as a function of the film’s thickness (fitted to the experimental data adjusting the hole sizes and porosity).

These results confirm that the microstructure and porosity of the film changed with the thickness. The fitted porosity steadily increased up to a thickness of 450 nm, from π = 20% for 100 nm to 62% of the bulk for 450 nm. Such high values of porosity were also obtained for thinner Si films prepared by GLAD [43]. This porous architecture produced in GLAD films is linked to the increase in the average distance between columns and the vanishing of other columns due to the shadowing effect (typical phenomenon driven by the growth competition, which is inherent to the GLAD process). For thicker films, the porosity still increased, but to a lesser extent, from π = 65% at for 600 nm to 69% of the bulk for 1000 nm. This evolution can be compared with densities measured for thick TiO_2_ films (>1 µm), which became uniform with the thickness [48], presumably related to the reduction of the column growth competition and their extinction.

### 3.2. Second Series: Sputtering Pressure from 2.5 × 10^−3^ to 15 × 10^−3^ mbar

#### 3.2.1. Morphology and Structure

The columnar microstructure and surface morphology of 400 nm thick W thin films were also influenced by the argon sputtering pressure, as shown in Figure 6.

For the lowest pressures (2.5 × 10^−3^ to 4.0 × 10^−3^ mbar in Figure 6a–6c, respectively), films exhibited a similar surface morphology, i.e., an elongated-shape of the columnar cross-section in the direction perpendicular to the particle flux (*y*-axis). Such anisotropic microstructure became even more marked as the argon sputtering pressure reduced with column widths reaching more than 500 nm for the largest spaces between columns of a few hundred nanometers in the *x*-direction. It is also interesting to note that this range of pressures gave rise to the highest column angle with *β* around 39° ± 2°, which was lower than the substrate angle α = 80°, as expected in GLAD deposition. From the top-view images, the column apex in front of the particle flux exhibited a smoother and more abrupt edge than the opposite side (in the shadowing region), which showed a serrated edge and a fibrous feature. This difference of morphological microstructure between the opposite sides of the columns is related to the ballistic character of W sputtered atoms. For the lowest argon sputtering pressures, W atoms have a ballistic behavior. The compact part on the column side facing the flux is due to the energy transfer of W atoms. They impinged on the column apex and were abruptly stopped leading to a dense material. On the opposite side, located in the shadowing zone, a few parts of W atoms arrived with a grazing incidence. They hit the column apex close to the shadowing zone and induced atomic mobility processes in the direction of the particle flux [36]. Columns were then formed by a dense zone on one side, whereas a fibrous and porous one was produced on the opposite side.

Increasing the argon sputtering pressure up to 15 × 10^−3^ mbar (Figure 6f) led to a less anisotropic microstructure. Columns showed a more isotropic cross-section, and voids between columns reduced but were still relevant. In addition, the column angle decreased and reached 18° ± 2° for this highest argon sputtering pressure. This change in microstructure has to be related to the thermalization of sputtered W atoms. Their energy decreased and the flux became scattered as the pressure rose. Taking into account our target-to-substrate distance (65 mm) and assuming that the calculated W atoms mean free path decreased from 7.2 cm down to 1.2 cm as the argon sputtering pressure changed from 2.5 × 10^−3^ to 15 × 10^−3^ mbar, respectively [49], direction and energy of W atoms impinging on the growing film were both modified. As determined by Westwood [50], the number of collisions required to thermalize a sputtered particle in an argon plasma is about 10. One can claim that for our range of pressures, W sputtered atoms changed from a ballistic to a thermalized characteristic mainly. From Barranco et al. [51], the thermalization degree *Ξ* and deposition angle *α* allow defining a microstructure phase map illustrating different kinds of microstructures in sputter-deposited thin films. A γ- to δ-type microstructural evolution is suggested for our W thin films as the argon sputtering pressure rose, i.e., some well-defined and isolated tilted columns at low thermalization degree (directional particle flux and low pressure), whereas a vertical and coalescent column-like structure with a high density of micro and mesopores occluded in the material for the highest one.

From XRD analyses (Figure 7), W films prepared at the lowest argon sputtering pressure (2.5 × 10^−3^ mbar) displayed better crystallinity.

An α and β phase mixture were clearly recorded with strong diffracted signals corresponding to α (110) and β (200) planes at 2*θ* = 47.12° and 41.76°, respectively. As previously reported in Section 3.1.1, it is interesting to note that the peak at 2*θ* = 47.12° related to the α phase was not symmetric with a shoulder on the low-angle side, which is assigned to the diffraction by the β (210) planes. Increasing the argon sputtering pressure gave rise to lower diffracted signals for both phases. The shoulder-peak previously noticed became even more substantial, and for pressures higher than 6.0 × 10^−3^ mbar, only signals connected to the β phase were clearly recorded. Peaks became broad and weak for the highest argon sputtering pressure of 15 × 10^−3^ mbar. As the pressure increased from 2.5 × 10^−3^ to 15 × 10^−3^ mbar, the α phase vanished and crystal size of the β phase reduced (from Scherrer formula and assuming β (200) peak) from 28 to 10 nm, respectively. This decrease in crystallinity with the vanishing of the α phase vs. pressure well agrees with previous studies reporting W thin film growth by conventional sputtering [49,52]. A high argon sputtering pressure increases the probability of collision between sputtered atoms traveling toward the substrate and argon atoms and ions (thermalization prevails). The mean free path of W sputtered atoms (well below the target-to-substrate distance of 65 mm), as well as their energy, is then reduced. As a result, the W particle flux tends to be less directional and less energetic. As suggested by Vüllers and Spolenak [49], the reduced energy of incoming W atoms favors low mobility adsorption. Other phenomena, such as implantation and adsorption of high mobility atoms which prevail at low working pressure, become negligible. In addition, migration of W atoms to the α phase growth sites is hindered, and the reduction of W mobility by adsorbed argon atoms is favored [53,54]. Since the formation of the β phase is often accompanied by a porous architecture, some growing defects, and a secondary growth phenomenon [42,49,55], the occurrence of such crystallographic structure is more sensitive to environmental interactions such as residual oxygen, which prevents the long-range order and development of α phase.

#### 3.2.2. Electrical Resistivity

Argon sputtering pressure did not only impact on the morphology and crystal structure of W GLAD thin films but it also influenced their electrical and anisotropic resistivity, as shown in Figure 8. A continuous and gradual increase in resistivity from ρ_300K_ = 5.7 × 10^−6^ to 5.6 × 10^−5^ Ω m was measured when the pressure changed from 2.5 × 10^−3^ to 15 × 10^−3^ mbar, respectively.

Whatever the argon sputtering pressure, the films’ resistivity was again two to three orders of magnitude higher than the W bulk value (ρ_300K_ = 5.4 × 10^−8^ Ω m [38]). This difference is commonly attributed to the grain boundary scattering of free electrons. XRD analyses showed a polycrystalline structure and the best crystallinity for films prepared with the lowest argon sputtering pressure of 2.5 × 10^−3^ mbar (Figure 7). For such a pressure, the coexistence of α and β phases was recorded with a crystal size of 22 and 23 nm, respectively (calculated from Scherrer formula and after deconvolution of diffracted signal related to β (210) and α (110) peaks). These values were higher than the electron mean free path in W, which is about 16 nm [56]. It means that the resistivity of GLAD W films produced at low argon sputtering pressure cannot be completely assigned to the electron scattering at the grain boundaries. The high voided microstructure (usually obtained in GLAD films) and large spaces between the tilted columns clearly observed from SEM images (Figure 6a) rather contributed to the high resistivity.

A substantial increase in resistivity was measured for an argon sputtering pressure higher than 6.0 × 10^−3^ mbar. This pressure range has to be correlated with the gradual evolution of the films’ microstructure (Figure 6), vanishing of the α phase and smooth decrease in the crystal size of the β phase down to a few nanometers for the highest pressure (Figure 7). Despite the shorter spaces between tilted columns as the pressure rose, voids still remained and contributed to the electron scattering. In addition, XRD patterns showed a broadening of diffracted signals related to the β phase, which means a crystal size lower than the electron mean free path. As a result, the film’s resistivity was dominated by characteristics of the remaining β phase, the latter being more resistive than the α phase.

A significant difference in electrical resistivity was measured following *x*- and *y*-directions (determined from the Bierwagen method [23]). This difference between *ρ_x_* and *ρ_y_* is once again related to the microstructure of W thin films. It was clearly demonstrated that GLAD deposition of columnar films with incident angles higher than 60° (critical-like angle) produced significant structural and uniaxial anisotropy in the substrate plane induced by the shadowing effect [57]. Thus, an anisotropic distribution of the grain boundary potential barrier heights was formed in the direction of the incoming atoms (*x*-axis), especially at low argon sputtering pressure due to the high directional behavior of the particle flux. W nuclei grew with a connection to each other by chains perpendicular to the direction of the shadowing effect (*y*-axis). The columnar growth exhibited an elliptical-shape cross-section, which became less marked as the pressure rises. Therefore, this difference of electrical resistivity in the orthogonal axes was less and less noticeable from 6.0 × 10^−3^ mbar, which is the pressure range corresponding to the formation of more isotropic cross-section and narrow columns. This change in microstructure is also related to the thermalization of sputtered W atoms (their energy decreases and the flux becomes less directional). Anisotropy was above 2.1 for the lowest argon sputtering pressures and progressively reduced to about 1.8 at 15 × 10^−3^ mbar. This *A**_ρ_* value was still high despite the isotropic columnar microstructure viewed from SEM pictures (Figure 6f). This means that electron scattering remained preferential in the direction to the particle flux (*x*-axis) and, thus a structural anisotropy remained as we can see in Figure 6e,f, where the density of columns was high, and they tended to keep bundled following the *x*-direction.

#### 3.2.3. Elastic Wave Propagation

The elastic properties of W films also depend on the argon sputtering pressure. Figure 9 illustrates the evolution of the group velocities and anisotropy of the pseudo-Rayleigh wave as the pressure changed from 2.5 × 10^−3^ to 15 × 10^−3^ mbar.

For the lowest pressures (2.5 × 10^−3^ to 4.0 × 10^−3^ mbar), the sputtered vapor was strongly directional, which favors shadowing conditions. The lateral growth allowed neighboring columns to touch and chain together, whereas shadowing prevented columns merging along the *x*-direction. This produced a preferential bundling of the columnar microstructure in the *y*-direction [13]. Hence, the velocity was significantly lower following the *x*-direction, which led to an important elastic anisotropic coefficient (*A_v_* = 1.8 to 2). Both velocities in the *x*- and *y*-directions increased with argon sputtering pressure. This evolution was correlated with the microstructural changes in the films and the reduction of porosity (Table 2).

Increasing the argon sputtering pressure up to 15 × 10^−3^ mbar (Figure 6f) gave rise to a less anisotropic microstructure. Columns showed a more isotropic cross-section and voids between them became smaller. As previously mentioned, this change in microstructure is related to the thermalization of sputtered W atoms. A further increase in pressure induced less inclined columns as the lack of vapor directionality effectively reproduced an isotropic deposition geometry. Nonetheless, a previous study [24] showed that the column tilt angle β has hardly any influence on velocities, and anisotropic velocity is mainly connected to the holes/columns geometry, especially their form factor. Due to the rather circular shape of the column cross-sections, *x* and *y*-velocities were similar with *v_x_* = 1800 m s^−1^ and *v_y_* = 2000 m s^−1,^ which means an anisotropy *A_v_* around 1.2. These results are consistent with an earlier publication where no anisotropy on the pseudo-Rayleigh waves propagating in gold films deposited by GLAD are reported (SEM observations similarly showed nearly circular column cross-sections) [46].

As reported in Table 2, the global porosity rapidly decreased as the argon sputtering pressure increased from π = 68 ± 10% for 2.5 × 10^−3^ mbar to 46 ± 10% of the bulk for 15 × 10^−3^ mbar. This result comes from the decrease in the inter-columnar space, leading to the densification of the film. Indeed, the high voided microstructure and large spaces between the tilted columns (obtained for the lowest pressures) clearly viewed by SEM analyses (Figure 6a–c) contributed to the global porosity. Although an increase in argon sputtering pressure usually leads to much more porous thin films prepared by conventional sputtering, this reverse trend has previously been reported in Liang et al. investigations for Mg films deposited by the GLAD method [58]. The authors also reached the same conclusion, i.e., a decrease in the film porosity due to shorter intercolumnar spaces obtained at high pressure.

## 4. Conclusions

Tungsten thin films were prepared by DC magnetron sputtering by the GLAD technique. A constant deposition angle α = 80° was used for all depositions. For a first series of films, a constant argon sputtering pressure of 4.0 × 10^−3^ mbar was used, whereas the film’s thickness was progressively changed from 50 to 1000 nm. An anisotropic microstructure was developed as the film’s thickness increased. The column angle was kept constant around 39° whatever the film’s thickness and columns became more and more asymmetric with elongated columnar cross-sections perpendicular to the incoming particle flux. This anisotropic microstructure was correlated with the ballistic character of sputtered particles (high directional flux), and the shadowing effect was effective from the early growing stage. DC electrical resistivity was progressively reduced as a function of the film’s thickness due to an increase in the grain size and the occurrence of the α-W phase. Resistivity, as well as surface elastic wave velocity, were reduced as the film’s thickness increases due to a more voided architecture (an increase in the micro- and nanoporosity). They also both exhibited an enhanced anisotropy up to 450 nm thick, which was correlated with a thickness range corresponding to a better-defined anisotropic microstructure.

For the second series, deposition time was adjusted to get a constant film thickness of 400 nm, and the argon sputtering pressure was systematically varied from 2.5 × 10^−3^ to 15 × 10^−3^ mbar. The elongated shape of the columnar cross-section produced with the lowest pressures in the direction perpendicular to the particle flux progressively transformed to less tilted and narrow columns with a more isotropic cross-section for the highest pressures. This morphological evolution was assigned to the predominance of thermalized tungsten atoms, more dispersive and less energetic tungsten atoms impinging on the growing films. Increasing the argon sputtering pressure, the α phase completely vanished, and films became poorly crystallized with the growth of a nano-crystallized β phase. This decrease in crystallinity was associated with an enhancement of the electrical resistivity, whereas surface elastic wave velocity was gradually increased due to the development of less spaced columns (decrease in porosity). The columnar microstructure became more homogeneous at high pressure and induced a more isotropic behavior of electronic conduction and elastic wave propagation in columnar tungsten thin films.

## Figures and Tables

**Figure 1 nanomaterials-10-00081-f001:**
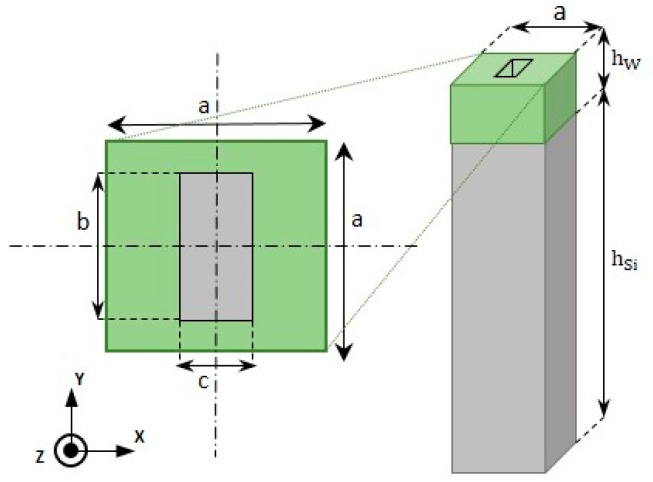
Scheme of the square cuboid unit cell (lattice constant *a*) composed of Si substrate (grey; thickness *h*_Si_) and tungsten film (green; thickness *h*_W_). Position and dimensions (length *c* and width *d*) of the parallepipedic hole dug in the tungsten film are indicated.

**Figure 2 nanomaterials-10-00081-f002:**
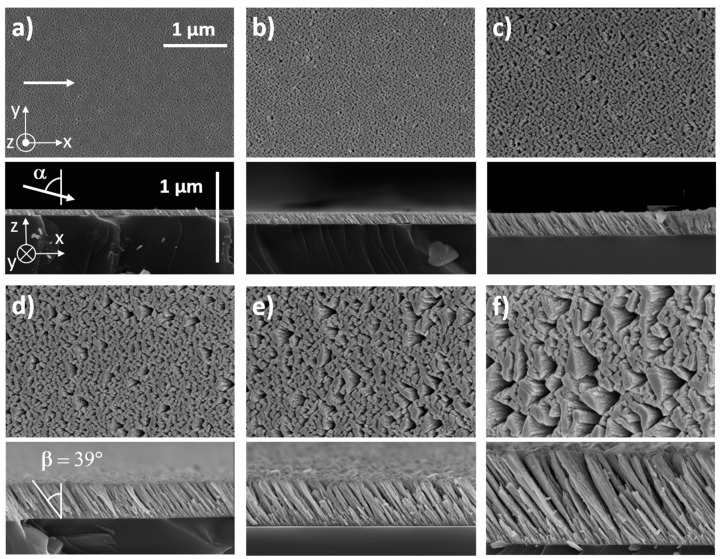
Top and cross-section views performed by SEM of W films sputter-deposited with a substrate angle α = 80° and an argon sputtering pressure *P_Ar_* = 4.0 × 10^−3^ mbar. White arrows indicate the direction of incoming particle flux. Column angle β was nearly constant (39° ± 2°) whatever the film’s thickness, which was systematically increased from: (**a**) 50; (**b**) 100; (**c**) 200; (**d**) 450; (**e**) 600, and (**f**) 1000 nm.

**Figure 3 nanomaterials-10-00081-f003:**
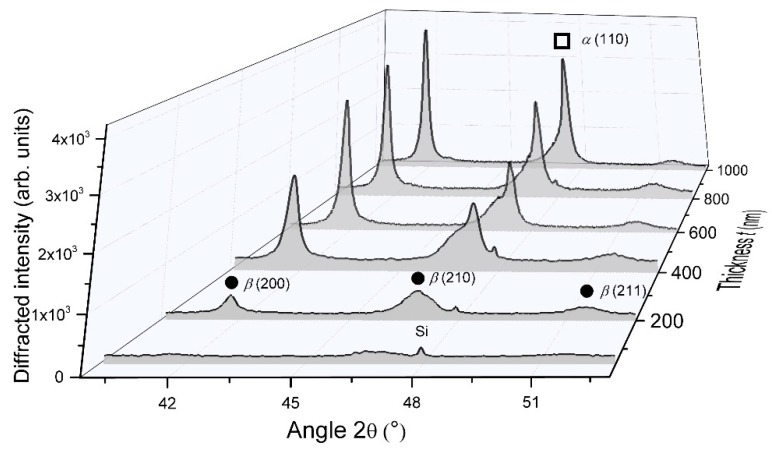
XRD patterns of W thin films sputter-deposited on (100) Si substrate with an angle α = 80°, an argon sputtering pressure of 4.0 × 10^−3^ mbar and for a thickness changing from 50 to 1000 nm. Only the 41°–52° range is presented since no significant peaks were measured elsewhere. Diffracted signals showed the occurrence of α and β phases (π and λ, respectively).

**Figure 4 nanomaterials-10-00081-f004:**
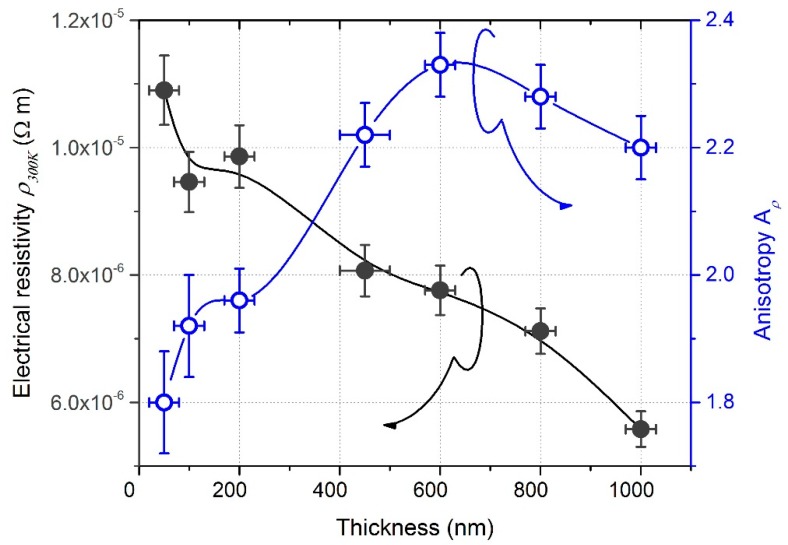
DC electrical resistivity and anisotropic resistivity measured at room temperature (300 K) as a function of the film’s thickness.

**Figure 5 nanomaterials-10-00081-f005:**
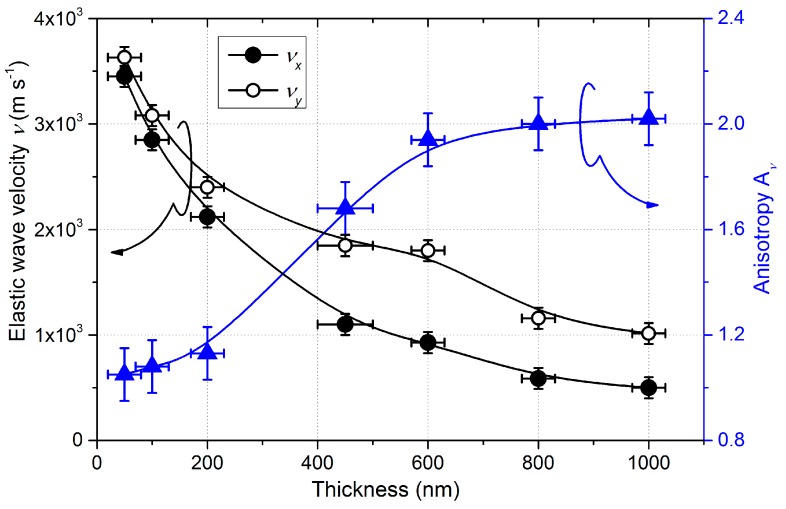
Group velocities of pseudo-Rayleigh waves along *x* and *y*-axes, and related anisotropic coefficient as a function of the W film’s thickness.

**Figure 6 nanomaterials-10-00081-f006:**
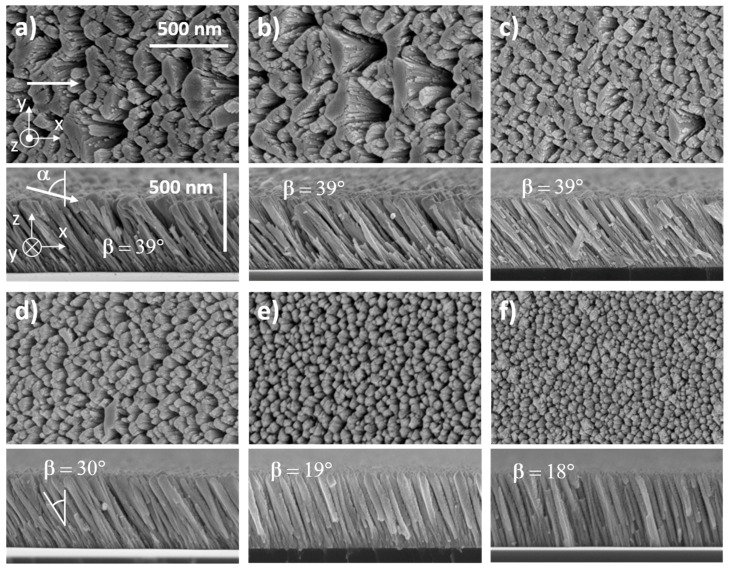
Top and cross-section observations by SEM of W films sputter-deposited with a substrate angle α = 80°. The deposition time was set to get a constant film thickness of 400 nm. White arrows indicate the direction of incoming particle flux. Column angle β changed from 39° ± 2° to 18° ± 2° as the argon sputtering pressure increased from 2.5 × 10^−3^ to 15 × 10^−3^ mbar: (**a**) 2.5; (**b**) 3.5; (**c**) 4.0; (**d**) 6.0; I 10, and (**f**) 15 × 10^−3^ mbar.

**Figure 7 nanomaterials-10-00081-f007:**
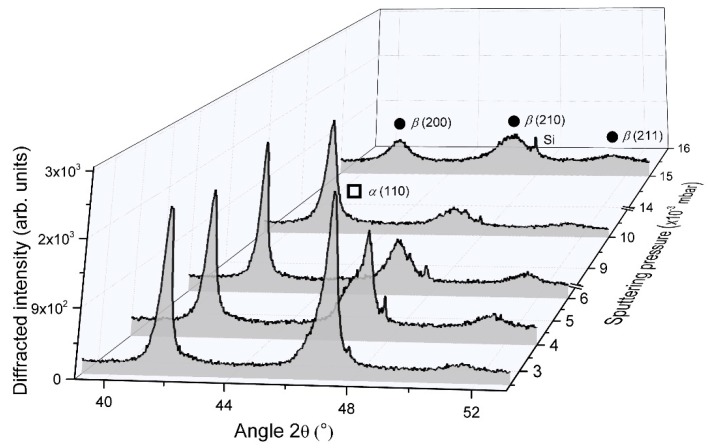
XRD patterns of 400 nm thick W thin films sputter-deposited on (100) Si substrate with an angle α = 80°, and for an argon sputtering pressure changing from 2.5 × 10^−3^ to 15 × 10^−3^ mbar. Diffracted signals show the occurrence of α and β phases (π and λ, respectively).

**Figure 8 nanomaterials-10-00081-f008:**
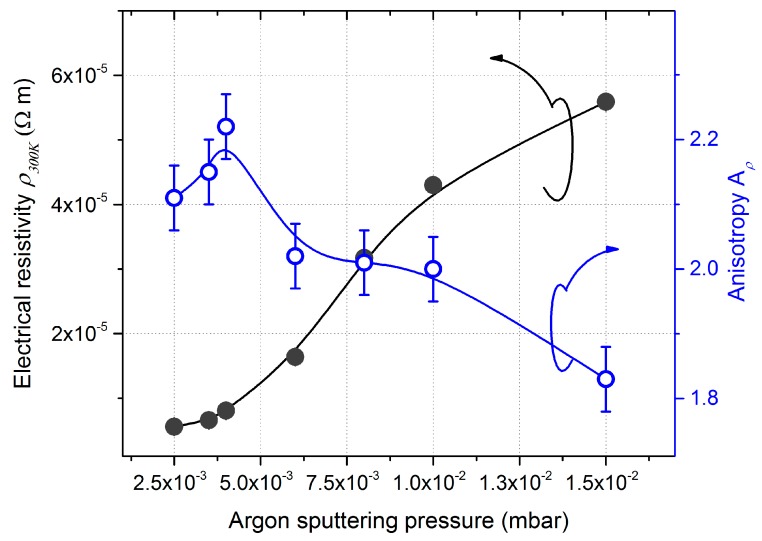
DC electrical resistivity and anisotropic resistivity measured at room temperature (300 K) vs. argon sputtering pressure of 400 nm thick W GLAD thin films prepared on a glass substrate with a deposition angle α = 80°.

**Figure 9 nanomaterials-10-00081-f009:**
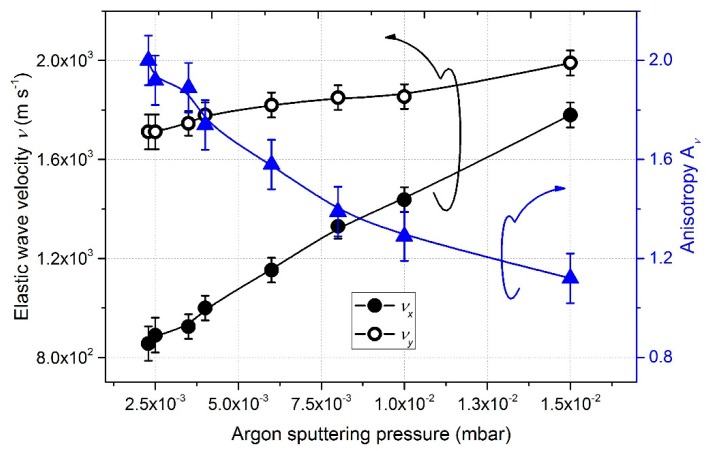
Group velocities of pseudo-Rayleigh waves along *x* and *y*-axes and related anisotropic coefficient as a function of the argon sputtering pressure.

**Table 1 nanomaterials-10-00081-t001:** Calculated global porosity as a function of the W film’s thickness.

Thickness (± 50 nm)	100	200	450	600	800	1000
Global porosity π (± 10% of the bulk)	20	33	62	65	66	69

**Table 2 nanomaterials-10-00081-t002:** Calculated porosity as a function of the argon sputtering pressure.

Pressure P_Ar_ (× 10^−3^ mbar)	2.5	3.5	4.0	6.0	8.0	10.0	15.0
Global porosity π (± 10% of the bulk)	68	65	62	59	55	53	46

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
