# Peer review of "Influence of Thickness and Sputtering Pressure on Electrical Resistivity and Elastic Wave Propagation in Oriented Columnar Tungsten Thin Films"

_nanomaterials, 2020, doi:10.3390/nano10010081_

Round 1
Reviewer 1 Report
The manuscript titled "Influence of thickness and sputtering pressure on 2 electrical resistivity and elastic wave propagation in 3 oriented columnar tungsten thin films" by Chargui et al. presents a somewhat straightforward, but nevertheless detailed study of the morphology as well as electrical and acoustic (as measured by elastic wave propagation) properties of W films grown by glancing-angle DC magnetron sputtering.
While the methods are well explained and the presented data are carefully obtained, I feel that the manuscript, in its current state, lacks in terms of research motivation, in particular as to why the authors decided to conduct this study on W films in particular. The way it reads, the work is just a systematic study with a series of characterization experiments performed just "because the authors could". As such, I strongly suggest the addition of a brief paragraph to the introducyion section of the manuscript, explaining why the study has been conducted on W films in more detail. Once this is done, I think the work would be suitable for publication in Nanomaterials.
Author Response
While the methods are well explained and the presented data are carefully obtained, I feel that the manuscript, in its current state, lacks in terms of research motivation, in particular as to why the authors decided to conduct this study on W films in particular. The way it reads, the work is just a systematic study with a series of characterization experiments performed just "because the authors could". As such, I strongly suggest the addition of a brief paragraph to the introducyion section of the manuscript, explaining why the study has been conducted on W films in more detail. Once this is done, I think the work would be suitable for publication in Nanomaterials.
As suggested by the Reviewer, the motivation of of conducting such a study has been added in the introduction part as follows:
"This study is motivated by the understanding of anisotropic behaviors in terms of electronic transport properties and elastic wave propagation in nanostructured thin films exhibiting a tilted columnar architecture. The choice of tungsten films is due to its ability to produce a columnar structure with tunable cross-section morphologies depending on deposition time and sputtering conditions."
Reviewer 2 Report
This work is very interesting and well prepared.
I have only a few minor remarks to make:
- The abstract in line 13 uses "W", it would be better to write Tungsten
- similarly in line 477 (summary)
- could be separated part cross-section and TOP view more clearly in Figures with SEM.
However, the scientific results are solid and well discussed.inline
Author Response
I have only a few minor remarks to make:
- The abstract in line 13 uses "W", it would be better to write Tungsten
- similarly in line 477 (summary)
- could be separated part cross-section and TOP view more clearly in Figures with SEM.
However, the scientific results are solid and well discussed.inline.
As suggested by the Reviewer, we wrote "Tungsten" instead of "W" in abstract and conclusion.
As suggested by the Reviewer, we separated top and cross-section views in Fig. 2 and 6.